

# The GRAS-2 Radio Occultation Mission

Joel Rasch[1], Anders Carlström[1], Jacob Christensen[1], Thomas Liljegren[1]

[1]Beyond Gravity Sweden AB, SE-40515 Göteborg, Sweden

*Correspondence to*: Joel Rasch (joel.rasch@beyondgravity.com)

**Abstract.** The second generation of the Global navigation satellite system Receiver for Atmospheric Sounding (GRAS-2) is a Radio Occultation (RO) instrument which is capable of providing 2000 atmospheric profiles per day. The instrument is hosted on all satellites in the Metop Second Generation (Metop-SG) series for polar orbit operation. The GRAS-2 instruments provide occultation measurements from the Galileo, GPS and BeiDou satellites at their common frequencies centred at 1575.42 MHz (L1) and 1176.45 MHz (L5). Using high-gain antennas and an ultra-stable oscillator, neutral bending angles are measured at an unprecedented accuracy of 0.3-0.4 µrad which is better than the requirement of less than 0.5 µrad. The RO signal will be measured deep into the troposphere using a novel open loop tracking scheme utilizing multiple correlator outputs for operation with a tailored ground processing algorithm optimized for extracting signals with low amplitudes approaching the noise floor limitation. Ionosphere measurements to an altitude of 600 km are also acquired.

## 1 Introduction

MetOp-SG is the follow-on to the current, first generation, series of MetOp satellites, which is a cornerstone of the global network of meteorological satellites. MetOp-SG will ensure the continuity of essential meteorological observations, improve the accuracy and resolution of the measurements, and also add new measurements. The MetOp-SG Programme is being implemented by ESA in collaboration with EUMETSAT. ESA is developing the prototype MetOp-SG satellites, including most of the associated instruments, and is procuring, on behalf of EUMETSAT, the recurrent satellites. Airbus Defence and Space (ADS) is the prime contractor for the development and production of the MetOp-SG satellites and leads a European industrial consortium including the entities responsible for the development of instruments. Beyond Gravity (former RUAG Space) develops the Radio Occultation (RO) instrument. The MetOp-SG will consist of two series of satellites (Sat-A and Sat-B), with three satellites of each series. This mission will provide continuous operation from polar orbit for more than 20 years. The RO instrument will be embarked on both satellites A and B. The RO mission primary objectives are to provide temperature and water vapour profiles. The RO measurements will also be used to derive ionospheric information, the tropopause height, the height of planetary boundary layers and surface pressure.

The Radio Occultation (RO) instrument tracks Global Navigation Satellite System (GNSS) signals from three different satellite constellations: the Global Positioning System (GPS), the Galileo system, and the BeiDou Navigation Satellite 2 System. The



RO instrument is also known as GRAS-2 since it builds on experience gained by its successful predecessor GRAS (GNSS Receiver for Atmospheric Sounding) (Bonnedal et al., 2010) on the MetOp satellites. The GRAS-2 RO instrument will provide about 2000 occultation measurements per day, thanks to simultaneous tracking of Galileo, GPS and BeiDou satellites. The instrument also has capacity to support a 4th constellation in a possible future upgrade. The mitigation of radio frequency interference from other transmitters in the GNSS bands has been emphasized during the development and a dedicated digital

filtering device has been developed for interference rejection. This paper presents the RO instrument design and its main performance parameters.

Radio occultation (RO) techniques have been used since the 1960s to obtain information of planetary atmospheres (Fjeldbo et al. 1971, Eshleman 1973). Occultation literally means the state of becoming hidden or disappearing from view. The occultation

technique consists of measuring the changes to a signal as the space vehicle passes in to or out of the shadow of a planet. For probing the atmosphere of the other planets in the solar system a radio signal is transmitted from Earth towards a planet and a space vehicle measures the apparent Doppler shift of this signal as it passes through various depths of the planets' atmosphere. The atmosphere causes bending or diffraction of the signal, which causes the frequency of the signal to be measured as different from the nominal as the receiver moves through the wave field. This is what is meant by the Doppler shift in the context of

occultations. The frequency stability of the transmitted signal as well as the receiver determines the accuracy of the measurement. The technique described above has a relatively low accuracy but given the lack of other measurement techniques for the other planetary atmospheres it still yields valuable data. Until the deployment of global navigation satellite systems (GNSS) such as GPS and Galileo the RO technique was of little interest for the Earths' atmosphere. The GNSS provides high accuracy signals and enabled global scale monitoring of tropospheric/stratospheric temperature, pressure and humidity profiles

with high accuracy and vertical resolution (Kursinski et al., 2000, Rocken et al. 1997, Yunck et al., 2000). In addition, ionospheric electron density profiles and scintillation properties are obtained (Hajj and Romans 1998, Schreiner et. al., 1999, Sokolovskiy et. al., 2002, Hocke and Igarashi 2002). The GNSS RO technique is different from occultation measurements of the other planets in that the signal is received by a Low Earth Orbit (LEO) satellite as it passes in or out of the Earths' shadow. The LEO satellite completes an orbit in around 100 minutes, whereas the GPS and Galileo satellites complete an orbit in 12

and 14 hours respectively. This means that an occultation takes place during around 5 minutes, and the GNSS is relatively still during this time. In turn this means that a vertical column of the atmosphere is sounded over one geographical position during a time frame over which the atmosphere changes very little. This data is very useful and routinely used for Numerical Weather Prognosis (NWP) (Healy and Thépaut 2006, Cucurull et. al. 2007, Aparicico and Deblonde 2008). The relative impact of the data is especially high in the stratosphere.


On MetOp-SG the signals of the occulting satellites are received via two occultation antennas: facing the satellite velocity direction, and the anti-velocity direction, respectively, see Fig. 1. The instrument also receives GNSS signals via a third zenith pointing antenna with a wide conical coverage. It acquires and tracks GNSS signals and provides the associated observables



as part of its measurement data to the ground segment to be used to compute the precise orbit of the spacecraft as well as the offset of the receiver clock. In addition, real-time navigation results are used by the instrument to control its operation. Dual-frequency zenith antenna measurements are also used for measuring electron content of the ionosphere above the orbit altitude.

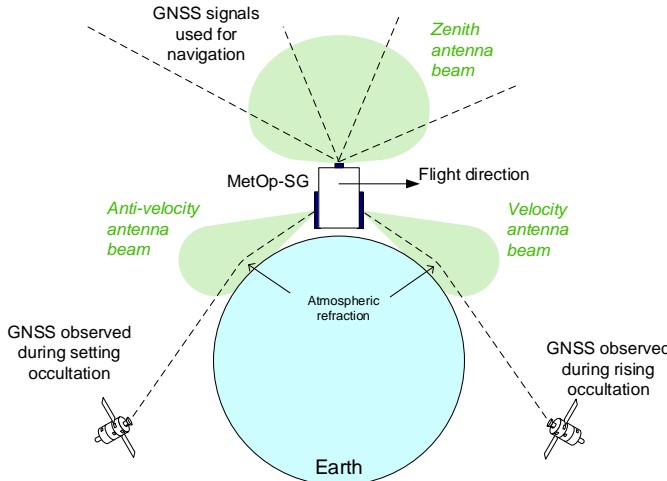

**Figure 1: Radio Occultation conceptual geometry. The field-of-view of the different antennas are illustrated in green.**

## 2 Instrument design

The instrument consists of seven units mounted on three faces of the MetOp-SG satellite as shown in the Fig. 2.

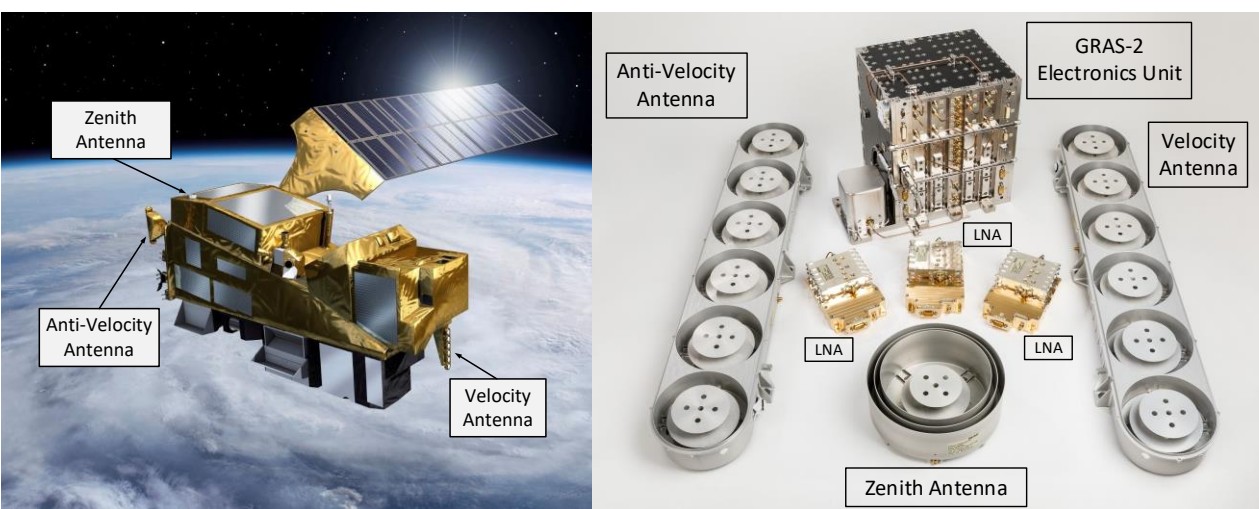

**Figure 2: MetOp-SG satellite (left, courtesy of ESA) and Radio Occultation instrument (right). The RO instrument includes: GRAS-2 Electronics Unit (GEU), Anti-Velocity Antenna, Velocity Antenna, Zenith Antenna, three Low Noise Amplifiers (LNA) for local signal amplification at each antenna.**




During the design phase, a trade-off analysis was made concerning the addition of redundant modules in the electronics to optimise the overall reliability for the 7.5 year nominal mission duration. This resulted in a design concept with partial redundancy where the power, interface, and navigation functions have redundant modules, at the expense of some additional mass as compared to a non-redundant instrument, see Table 1 below.


| | | Partially redundant concept for MetOp-SG | Non-redundant concept for comparison |
|---|---|---|---|
| **Mass:** | GRAS-2 Electronics Unit | 14 kg | 10 kg |
| | Antennas and LNAs | 8 kg | 8 kg |
| **Power:** | Power Bus Voltage | 50 V | 50 V |
| | Power consumption | 41 W | 41 W |
| **TC/TM interface:** | Electrical | SpaceWire | SpaceWire |
| | Packet format | PUS/CCSDS | PUS/CCSDS |
| | Data rate (orbit average) | 1 Mbps | 1 Mbps |
| **Reliability:** | Reliability over 7.5 years | 0.85 | 0.79 |

**Table 1: RO Instrument Accommodation Characteristics**

The main instrument performance characteristics are presented in Table 2 below.


| | Performance | Remark |
|---|---|---|
| **Number of occultations** | 1900-2100/day | with 27-30 satellites per GNSS constellation |
| **Bending angle accuracy** | <0.5 urad | >35 km altitude |
| **Altitude coverage** | Surface to 600 km | Adjustable |
| **Sample rate** | 200/250 Hz | GPS and BeiDou / Galileo |

**Table 2: RO Instrument Performance Characteristics**

The instrument block diagram is shown in Fig. 3, and the design is described in the following sections.



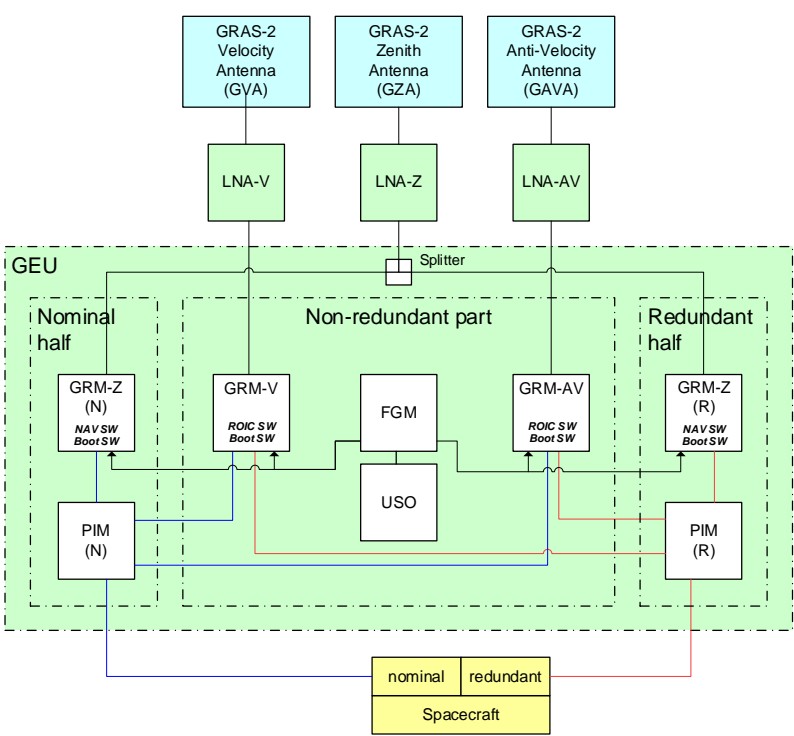


**Figure 3: Instrument block diagram.**

## 2.1 Antennas

The GRAS-2 Zenith Antenna (GZA) was developed in the Sentinel 1/2/3 programme. It consists of a patch-excited cup with

choke rings for optimum decoupling from the surrounding satellite structure. The antenna receives right-hand circularly polarized signals in two frequency bands denoted L1 (1575.42 MHz +/- 10.23 MHz) and L5 (1176.45 MHz +/- 10.23 MHz). Both signals are output through a single coaxial connector (SMA).

The velocity and anti-velocity antennas (GVA and GAVA) are identical. The design is an array antenna with the same type of

element as for GZA but the size of the cup is reduced to obtain an appropriate inter-element separation distance. A separate beam-forming network in planar technology is placed at the rear side of the antenna. A broadband fixed beam steering is obtained by a linear variation of the length of the lines to the antenna elements. The antenna gain patterns at the respective centre frequency are shown in Fig. 4.





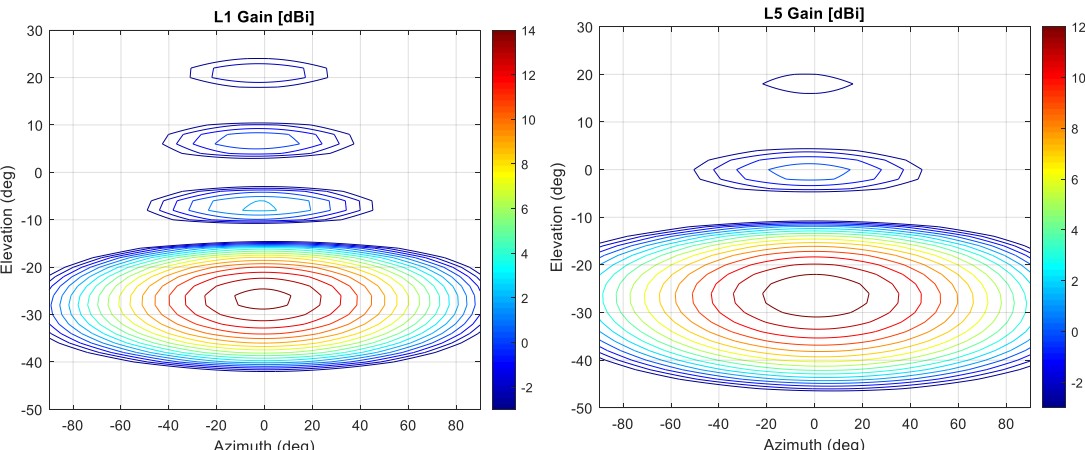

**Figure 4: RO antenna gain patterns for the L1 (left) and L5 (right) signals. The colour scale shows antenna gain in [dBi].**

## 2.2 Low Noise Amplifier (LNA)

The LNA is used for filtering and amplifying the weak received RF signal from the antenna. For interference protection, two band-pass filters centred around the L1 and L5 frequencies are placed at the LNA input. The interference rejection of the coaxial cavity filters is more than 50 dB at +/- 100 MHz from the respective centre frequency. The amplifier stages are designed around discrete Gallium Arsenide (GaAs) low-noise metal–semiconductor field-effect transistors. The achieved gain is more than 35 dB and the typical measured noise figure of the complete LNA unit is 1.6 dB at L1 and 2.0 dB at L5. The reason for the difference is that a higher filter order is used at L5 for protection against radar interference in the 1215-1350 MHz band (>35 dB rejection).

## 2.3 GRAS-2 Electronics Unit (GEU)

The GEU includes the following modules:

- **GNSS Receiver Module (GRM)**
  The amplified signal from each LNA is connected to a dedicated GRM. It contains signal amplification, down-conversion, filtering, analogue-to-digital conversion, digital signal processing, and a processor for the instrument control and processing functions needed in the respective GRM. The digital functions are contained in the ESA-developed *AGGA-4* device providing 36 signal channels. Signal tracking is performed by software-controlled algorithms. The GRM packets the measurement data for further distribution over a SpaceWire serial link to a router in the Power and Interface Module (PIM).

- **Frequency Generator Module (FGM)**
  The purpose of the FGM is to provide local oscillator (LO) signals with low phase noise for the down-conversion of the GNSS signals within the GRMs. The LO signals are phase-locked to an input reference signal at 10 MHz from



the Ultra-Stable Oscillator (USO). The reference signal is also amplified and output to the backplane for distribution to all GRMs within the GEU.

- **Ultra-Stable Oscillator (USO)**
The USO provides the 10 MHz reference signal to the FGM. The USO is mounted on an extended GEU baseplate next to the other modules. It includes a mu-metal housing for protection against magnetic field variations, which would otherwise affect the frequency of the oven-controlled crystal oscillator. The short-term stability of the USO is better than 0.5e-13 for gate times in the range 1s to 100s. This frequency stability enables very accurate measurements of the Doppler shift introduced by the atmospheric refraction of the radio occultation signals.

- **Power and Interface Module (PIM)**
The Power and Interface Module (PIM) receives primary power from the spacecraft, converts it to secondary voltages and distributes these to the different modules within the GEU and to the externally located LNA units. The PIM also provides the communication interface towards the spacecraft through SpaceWire links from the GRM boards connected to a SpaceWire router hosted in PIM.

## 3 Interference mitigation

The L5 GNSS band is shared with aviation navigation systems known as Distance Measuring Equipment/Tactical Air Navigation (DME/TACAN). If not accounted for, the transmitted signals from these systems can cause significant degradation of the RO measurements. Hence, the GRMs used for the velocity and anti-velocity receiver chains include Frequency Domain Adaptive Filtering (FDAF) processing for interference suppression. The FDAF processing is performed in a dedicated integrated circuit named *Frodo* before the signal is input to the *AGGA-4* device.


The *Frodo* device provides a configurable adaptive notching of narrow frequency bands inside the receive signal bandwidth with an updating interval of 8 microseconds. The interference mitigation capability has been tested for a worst case DME/TACAN scenario with 400 simultaneous interference sources consisting of pulsed transmitters, each with a bandwidth of 125 kHz, spread over a 30 MHz band with 1 MHz spacing. This corresponds to an Interference-to-Signal ratio (I/S) of about
60 dB. The performance is characterised by comparing the SNR of the useful signal for the cases with/without interference, and with/without FDAF processing. The test results show that the worst SNR loss due to interference is reduced from 17 dB to 3 dB when the FDAF processing function is activated. The Frodo also provides monitoring through several telemetry parameters including "total RF input power", "input power spectrum", and "fraction of spectrum being jammed". The characteristics of the interference mitigation function are summarised in Table 3.






| Frequency resolution | 120 kHz |
|---|---|
| Time resolution | 10 us |
| Total processing bandwidth | 28 MHz |
| Interference-to-Signal Ratio | Up to 60 dB |
| SNR loss at maximum interference | <4 dB  (a typical measured value is 3.4 dB) |

**Table 3. Interference Mitigation Device Characteristics**

## 4 Tracking scheme and simulated instrument performance optimization

An Instrument Data Simulator (IDS) has been developed to simulate realistic instrument data. An important part of the IDS for the GRAS-2 instrument is the Wave Optics Propagator (WOP) developed in a collaboration between Chalmers University of Technology and Beyond Gravity (Rasch 2014). This WOP was also used to generate the atmospheric modulation for the signals used for instrument testing. The IDS generates realistic measurement data and has been used extensively to fine-tune the tracking algorithms during the development of the instrument. To describe the overall tracking scheme, it is best to use the concept of Straight Line Tangent Altitude (SLTA). It is the height above ground for a straight line drawn between GNSS and LEO at the point where the line is perpendicular to a radius drawn from the centre of curvature of the Earth. Above the troposphere the measure is more or less the same as the tangent altitude of the signal (i.e. the point where the signal makes its closest approach to the Earth surface), but deeper down in the atmosphere the measures start to diverge.  The overall scheme is as follows: above 80 km closed loop tracking of the Pilot components of L1 and L5 is performed; below 80 km's the Data component is tracked as well. The pilot and data signals are combined in postprocessing to yield a higher signal to noise ratio than the individual signals. The closed loop tracking uses loops to track both the carrier and code of the signals. When the signal goes into the troposphere the dynamics of the signals can become hard to follow. For this reason, an open loop tracking algorithm has been developed (Carlström et al. 2012), which ensures that all signal information throughout the atmosphere is made available for ground processing without gaps. The open loop tracking uses a range model to steer the carrier frequency of the replica signal. The code is tracked using a loop, but before code lock is achieved, and during gaps where code lock is lost, the range model is used also here. For setting occultations the open loop tracking starts at 20 km SLTA and ends at -300 km SLTA. It uses 5 correlators. For rising occultations open loop tracking starts at -300 km SLTA with 10 correlators. At -38 km SLTA it switches to 5 correlators. The large number of correlators allows the signal to be reconstructed in postprocessing even for the segments where there is no code lock. For an ideal atmosphere there would not be any signal deeper than -100 km, but due to the presence of ducts and the super-refractive phenomena it is expected that interesting features may appear even as far down as -300 km SLTA (Sokolovskiy 2014). All the heights mentioned above are configurable. Although extensive tuning has been performed with simulations it is likely that these heights will be changed after the satellites are launched. When producing the bending angle curve one may choose freely to use purely open or closed loop tracking data.

A comparison between the performance of GRAS and GRAS-2 is seen in Table 4.



|  | **GRAS on MetOp** | **GRAS-2 on MetOp-SG** |
|---|---|---|
| Number of instruments | 3 (1 per satellite) | 6 (1 per satellite) |
| Number of constellations | 1 (GPS) | 3-4 (Galileo, GPS, BeiDou, resources for one more) |
| Number of occultations | ~650 per day per instrument | ~2000 per day per instrument |
| Bending angle [rms] | 0.6 µrad | 0.5 µrad |
| Altitude coverage | 0-80 km, 80-300 km | 0-80 km, 80-600 km |
| Code tracking | Closed loop | Open and closed loop |
| Carrier tracking | Open and closed loop | Open and closed loop |
| USO Allan deviation | 1e-12 | 5e-13 |
| Reliability | 0.8 over 5 years | 0.85 over 7.5 years |

**Table 4. Comparison between GRAS and GRAS-2.**

## 5 Ground processor prototype

A Ground Processor Prototype (GPP) has been developed to evaluate and optimize the end-to-end performance of the instrument. As explained in Section 4, both open and closed loop tracking of the pilot and data signals are performed by the instrument. The quality of the final bending angle depends on how these signals are processed in the GPP. Especially for the open loop data in the rising occultations special algorithms are needed. In Fig. 5, the amplitude of 9 correlators (a 10[th] correlator is spaced a bit away from the others and used to track the noise level) for a rising GPS occultation is shown as obtained from IDS.





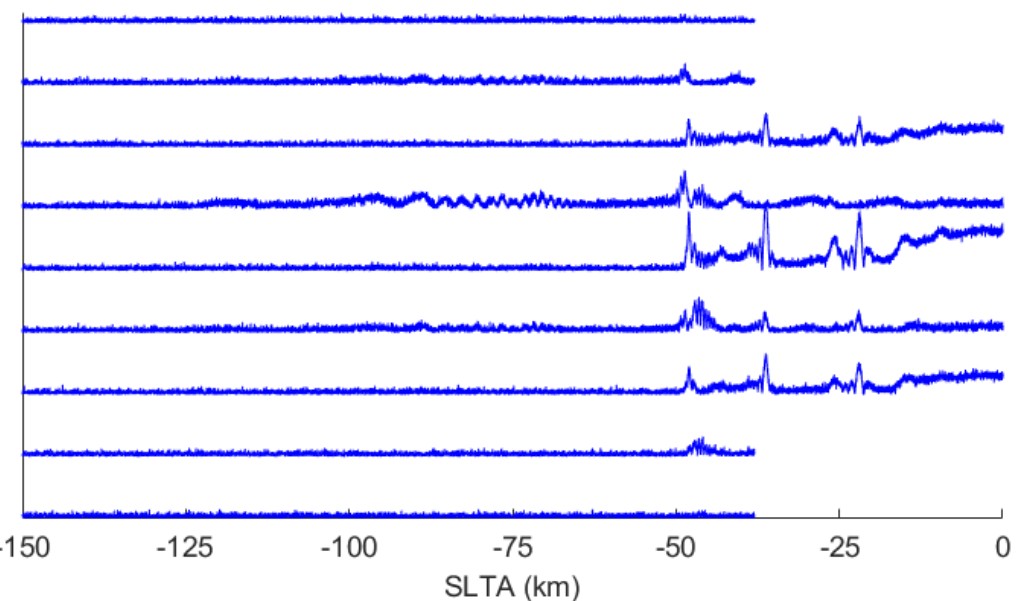

**Figure 5: Raw multiple correlator amplitudes (linear units) for a rising GPS L1C occultation simulated in the IDS. Correlator**
195   **numbering is 1 through 9 from bottom to top.**

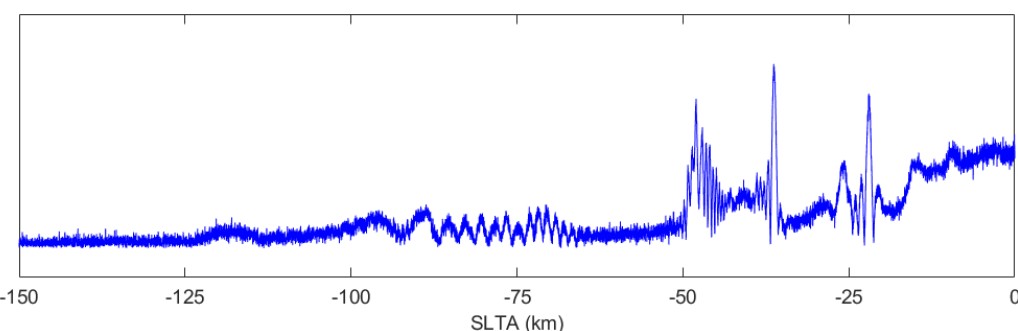

**Figure 6: Correlator amplitudes (linear units) after application of the peak correlator algorithm for a rising GPS L1C occultation**
**simulated in the IDS.**

200

When code lock is achieved the code loop adjusts the delay for the signal replica so that the centre correlator ($5^{th}$ from bottom)
is aligned with the auto-correlation peak of the signal and hence receives the maximum energy. Before code lock is achieved
it is impossible to know beforehand where this peak will be due to an unpredictable ionospheric delay to the signal. In the case
in Fig. 5, before code lock, the main signal energy is recorded by correlator 6. At the point where code lock is achieved, around
205   -50 km SLTA in Fig. 5, the loop adjusts itself rather quickly. This adjustment can be observed by the recorded time stamps
and code phase in the data, which means that in post processing we can estimate this ionospheric offset. Using a weighted sum





over correlators we can therefore calculate what the amplitude would have been for a fictitious correlator that is aligned with the peak all the way through the occultation. Such a peak correlator algorithm has been specially designed for this instrument concept. The resulting amplitude after this algorithm has been applied is seen in Fig. 6. Note also in the figure that the number of correlators drop from 9 to 5 at -38 km SLTA where they are no longer needed. The 9 correlators allow the code loop to find the peak and start tracking even for quite large ionospheric delays of ~100 m. The peak correlator algorithm allows us to find signal features for very deep occultations, which occur due to rather dynamic tropospheric conditions. To process the signal to bending angle various algorithms can be employed. In the stratosphere and above, the so-called geometrical optics (Fjeldbo et al. 1971) method is generally used, due to its ease of implementation and speed of execution. To resolve the multipath phenomena (Sokolovskiy 2001) that occur in the troposphere one needs to use more sophisticated methods such as the Canonical Transform (Gorbunov 2002) and Full Spectrum Inversion (Jensen et al. 2003). In the GPP we use the Phase Matching (PM) method (Jensen et. al., 2004). The method is based on stationary phase integrals and relatively straight-forward to use. The downside of the method is the rather heavy computations needed. Careful optimization has been performed for the GPP so that it can operate in real time (i.e. process an orbit worth of occultations in the time-frame of one orbit). Before the bending angle is computed, there are several more steps done by the GPP. The open and closed loop data is combined, where the open loop data is used in the low atmosphere and closed loop in the high atmosphere. In the transition region where both tracking methods are used a weighted average is calculated. The final step to calculate the neutral bending angle is to remove the ionospheric bending by combining the L1 and L5 data (Vorob'ev and Krasil'nikova 2002). The GPP is designed to be used for instrument development, at instrument testing, at satellite level testing, and for the satellite in orbit verification phase after the satellite has been launched.

## 6 RO instrument performance

The low-noise performance of the instrument provides a measured root-mean-square (RMS) accuracy of the post-processed bending angle measurements of 0.3-0.4 µrad (typical values) in the upper atmosphere, see Fig. 7, which is better than the EUMETSAT requirement of 0.5 µrad. This is a measured result which is obtained by a test set-up which includes a signal generator capable of simulating the three GNSS constellations (GPS, Galileo, BeiDou) and the modulation from a representative set of atmospheric profiles. The modulation profiles used by the signal generator was generated using realistic atmospheric refractivity profiles, and the WOP (see Section 4). The raw instrument output is provided to the GPP which produces the neutral bending angle profiles. The result below show that we expect no significant difference in performance between the three GNSS constellations.





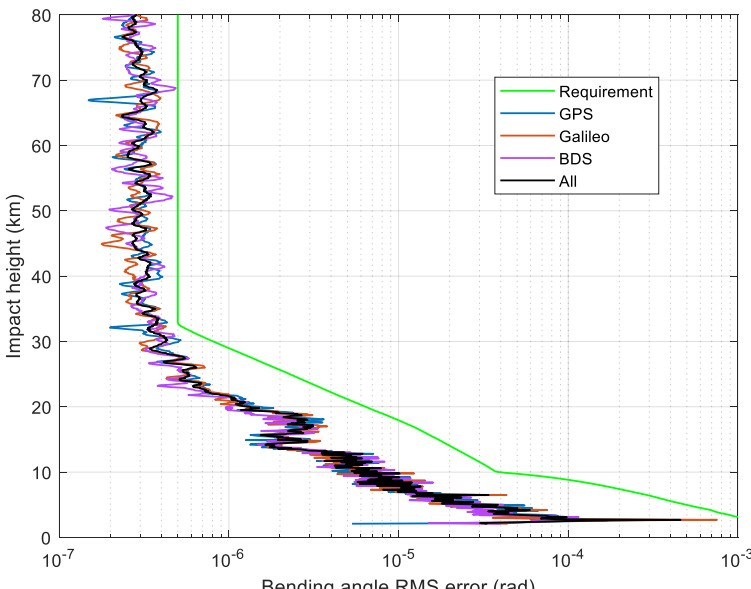

**Figure 7: Measured neutral bending angle errors.**

## 7 Satellite level integration and test

The first RO instrument, the Proto-Flight Model (PFM), was successfully delivered to Airbus Defence and Space in 2020.
Figure 7 shows the PFM instrument integrated on the Sat-A PFM in Toulouse. To enable accurate and repeatable testing at satellite level, antenna test-caps are mounted over the antennas when the instrument is integrated on the satellite. A dedicated test equipment then provides the stimuli signals to each antenna such that the function and performance of the instrument can be tested in several stages of the satellite test campaign including also thermal vacuum conditions.

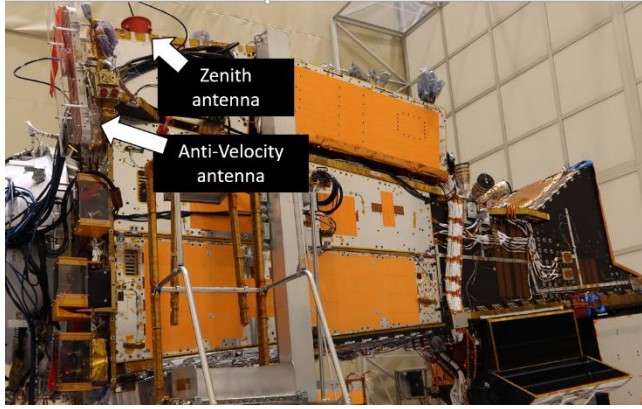

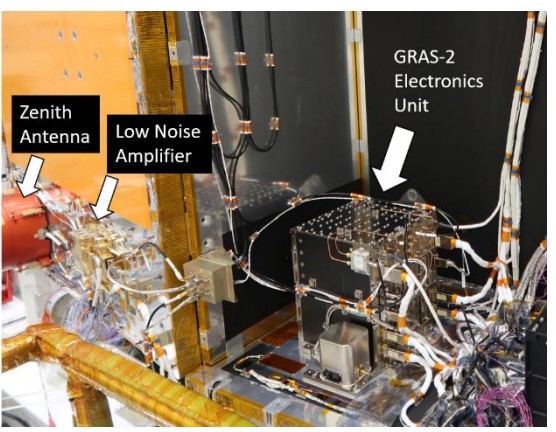



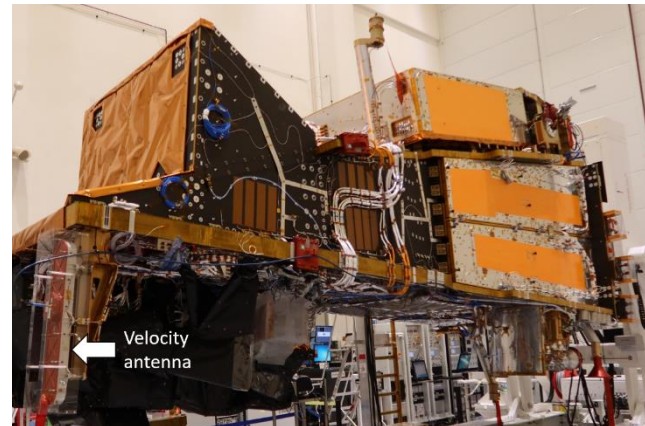
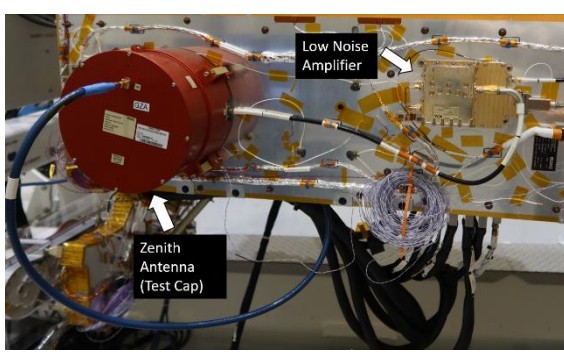

**Figure 7: Proto-flight model satellite and RO units with test-caps to allow testing on satellite level. Source: Airbus Defence and Space SAS/ GmbH. Notes: MetOp-SG is a collaborative programme between ESA and EUMETSAT. Airbus Defence and Space is Prime contractor, under ESA, for the two Satellites series contracts. Picture shows satellite in assembly phase and not in its final configuration**

## 8 Conclusion

The RO instrument on Metop-SG is designed to provide 1900-2100 occultation measurements per day, thanks to simultaneous tracking of Galileo, GPS and BeiDou satellites. The instrument also has capacity to support a 4[th] constellation in a possible future upgrade. The mitigation of radio frequency interference from other transmitters in the GNSS bands has been emphasized during the development. A dedicated digital processing device has been developed for this purpose. All performance critical parameters, including antenna gain, amplifier noise figure, and oscillator stability are superior to those of current generation
instruments. Hence, the measured root-mean-square (RMS) accuracy of post-processed bending angle measurements in the upper atmosphere is typically in the range 0.3-0.4 microradians. The GRAS-2 instrument is an important contributor to numerical weather predictions, but it is also highly capable of ionospheric monitoring of electron density and scintillations, as well as for research of phenomena related to deep occultations such as ducting and super-refraction. Novel ground processing algorithms have been developed to utilize the data recorded by multiple correlators providing optimum performance for deep
occultations.

## Author contributions

The work and results described in this paper has involved many people, but the authors have played key roles. The manuscript itself was prepared almost completely by Anders Carlström and Joel Rasch with support and advice from Thomas Liljegren and Jacob Christensen. The sections regarding instrument details and performance were prepared mainly by Anders, and the
parts on the simulations and ground processing mostly by Joel.



**Acknowledgments**

MetOp-SG is a collaborative programme between ESA and EUMETSAT. Airbus Defence and Space is Prime contractor, under ESA. Beyond Gravity is subcontractor to Airbus.

**Competing interests**

The contact author has declared that none of the authors has any competing interests.

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
