# Peer review of "The GRAS-2 Radio Occultation Mission"

_Atmospheric Measurement Techniques, 2024_

## Author Response (AR1)

**Response to referee #1**

**Referee comments**

General comments

A) The authors describe the new GRASS-2 instrument on the Metop Second Generation (Metop-SG) satellites. GRASS-2 will provide radio occultation (RO) measurements based on the Galileo, GPS and BeiDou navigation satellites. GRASS-2 is a further development of the GRASS receiver on the current Metop satellites. The paper is well structured and written with clear results and conclusions. My recommendation is to accept the paper.

Specific comments

B) From the paper ist is not clear when GRASS-2 data will be available.

C) Lines 57-58: NWP = Numerical Weather Prediction

D) Line 85, Table 2: What is the advantage of this (very) high sample rates (250 Hz for Galileo)?

**Author responses**

A) Thank you for your comments.

B) Data should become available sometime during the first half of 2026. We will add a sentence with this information in the manuscript.

C) NWP = Numerical Weather Prediction, yes you are quite right, we will correct this.

D) Regarding the sampling rate. Judging from COSMIC and GRAS sample rates of 50 and 100 Hz are sufficient to produce high quality RO data for NWP. Higher sample rates are mainly thought to become important in the troposphere where moisture can cause the received signal frequency to change very rapidly. Higher sample rates may contribute to improve the bending angle retrieval in the troposphere. The particular values for sampling rates are chosen so that they align with the

navigation data bits of the GNSS signals. The E1b signal has a symbol rate of 250 Hz. We will add some sentences in section 4 with this information.

**Changes in manuscript**

A) None

B) Sentence added on rows 26-27 in introduction.

C) Sentence corrected on row 63.

D) Some sentences were added on rows 190-194 in section 4.

**Response to referee #2**

**Referee comments**

General comments.

A) A high level presentation of the GRAS-2 instrument was provided, with some details on the instrument performance, in particular these were measured against its predecessor the GRAS instrument. The information is presented in a clear concise manner and only in a few instances there likely could be some additional qualifying information added to add the context and in some cases the motivation beyond the results presented.

Specific comments.

B) Around line 55 on page 2, the statement, "a vertical column is sounded over one geographical position" could be misleading to those unfamiliar with details of the technique. There is more detail later in the manuscript describing additional qualifications. It may be helpful to consider how this could be restated, "a vertical column is sounded along the path of the received signal". Apologies for not having a concrete suggestion for rewording but it may be worth another examination.

C) The SpaceWire router is mentioned a few times, and its significance may be lost. It may be worth prefacing the first mention of the SpaceWire router than the spacecraft includes a router compliant with the ESA SpireWire standards ... .

D) In the ground processing prototype section there is a bending angle product presented. Is this being requested to be included in the ground processing? Assume the data provided to EUMETSAT likely will be lower level prior to generation of bending angle? Please clearly state that the ground processing prototype also includes processing to bending angle (at the request of EUMETSAT?, and for the estimation of instrument error in bending angle?). An obvious motivation for signal processing to bending angle, is shown in the next

RO instrument performance section. Mentioning this as motivation in the previous section would make sense if you would agree, and tie these sections together more directly.

E) Lastly, the satellite integration and tests section was very sparse. Assuming Airbus provided the antenna test-caps and led the testing. Are any of the results from these measurements and tests included in this manuscript?

Technical Corrections

No major technical corrections were found, there are a few suggestions listed below:

F) Page 2 around line 39. "Occultation literally means …" is a little colloquial. Maybe: "Occultation in this case refers to the GRAS-2 receiver becoming hidden or disappearing from view of a signal transmitter."

G) Page 2 around line 54. "…, whereas the GPS and Galileo satellites … " consider being a bit more specific. "…, whereas the transmitting GNSS satellite (e.g. GPS or Galileo) complete …"

H) Page 2, line 55. Regarding the statement that the occulatation takes place in around 5 min. Suggest: "With the current positioning of the LEO receivers and the GNSS transmitters, an occulation takes place over around 5 min, …".

I) Page 13, at the end of the conclusions would consider adding another sentence to reprise the previous section (delivery to Airbus). "An initial instrument has been delivered to Airbus in preparation for delivery of the first of the MetOp-SG spacecraft" or something similar.

**Author responses**

A) Thank you for your comments.

B) Yes, this sentence could be misunderstood. We will rephrase it. We will update the sentence to "For each point in time, the signal path between the GNSS and LEO will be a slightly bent line. Most of the bending is generated when the path makes its closest approach to the Earth and the atmospheric refraction is

strongest. The atmospheric information contained in the bending angle of the RO signal for a given sample time is thus generated by a relatively small segment along the lowest part of the signal trajectory. The successive samples of the RO measurement thus form a vertical column of such segments. During the 5 minutes of the occultation, the atmospheric state inside this column will change very little."

C) Yes, we should say something more about the spacewire. We will add a sentence like you suggested.

D) Indeed, the motivation for the development of the GPP is to be able to assess the instrument error contribution to the bending angle error. We will add some text tying sections 5 and 6 together.

E) No. The test results that we present in the manuscript in section 6 are at instrument level, and as you noted there are no test results in section 7. At instrument level we use a high end GNSS signal generator that is directly fed to the LNA's in the instrument. At satellite integration level signals are generated by a playback unit and fed to the test caps and into the antennas. The playback unit and the test caps add noise to the bending angle curve and gives a result that is not representative of the instrument performance. So the purpose of the tests at satellite level are not to judge the performance of the instrument, but rather to ensure that the instrument is undamaged and has been installed correctly. That evaluation is done by comparing with reference measurements made by us using the test caps and the playback unit here in our lab as part of our instrument test campaign.

F) We can remove the sentence "Occultation literally means …". It is not important.

G) Yes, the text you suggest is good. We will use it.

H) Yes, we see your point. We will use a sentence similar to the one you suggest.

I) Yes, we will add a sentence reiterating the current state of the deliveries to Airbus.

**Changes in manuscript**

A) None.

B) A sentence deleted and several added on rows 55-62 in introduction.

C) Two sentences were added on rows 94-96 in section 2.

D) A few sentences were added on rows 200-204 in section 5.

E) None.

F) Sentence "Occultation literally means the state of becoming hidden or disappearing from view." removed on row 40 in introduction.

G) Sentence modified according to suggestion on row 55.

H) Sentence modified according to suggestion on row 55-56.

I) The sentence "All six instruments have now been delivered to Airbus and the first MetOp-SG satellite is foreseen to be launched in 2025." Was added to the conclusion (section 8).